# A Conditional Privacy Preserving Generalized Ring Signcryption Scheme for Micro Aerial Vehicles

**DOI:** 10.3390/mi13111926

**Published:** 2022-11-08

**Authors:** Insaf Ullah, Muhammad Asghar Khan, Ako Muhammad Abdullah, Syed Agha Hassnain Mohsan, Fazal Noor, Fahad Algarni, Nisreen Innab

**Affiliations:** 1Hamdard Institute of Engineering and Technology, Hamdard University, Islamabad 440000, Pakistan; 2Computer Science Department, College of Basic Education, University of Sulaimani, Sulaimaniyah 00964, Kurdistan Region, Iraq; 3Department of Information Technology, University College of Goizha, Sulaimaniyah 00964, Kurdistan Region, Iraq; 4Ocean College, Zhejiang University, Zheda Road 1, Zhoushan 316021, China; 5Faculty of Computer and Information Systems, Islamic University of Madinah, Madinah 400411, Saudi Arabia; 6College of Computing and Information Technology, The University of Bisha, Bisha 67714, Saudi Arabia; 7Department of Computer Science and Information Systems, College of Applied Sciences, AlMaarefa University, P.O. Box 71666, Riyadh 11597, Saudi Arabia

**Keywords:** micro aerial vehicles, security, signcryption, elliptic curve cryptography, ring signcryption

## Abstract

Micro Aerial Vehicles (MAVs) are a type of UAV that are both small and fully autonomous, making them ideal for both civilian and military applications. Modern MAVs can hover and navigate while carrying several sensors, operate over long distances, and send data to a portable base station. Despite their many benefits, MAVs often encounter obstacles due to limitations in the embedded system (such as memory, processing power, energy, etc.). Due to these obstacles and the use of open wireless communication channels, MAVs are vulnerable to a variety of cyber-physical attacks. Consequently, MAVs cannot execute complex cryptographic algorithms due to their limited computing power. In light of these considerations, this article proposes a conditional privacy-preserving generalized ring signcryption scheme for MAVs using an identity-based cryptosystem. Elliptic Curve Cryptography (ECC), with a key size of 160 bits, is used in the proposed scheme. The proposed scheme’s security robustness has been analyzed using the Random Oracle Model (ROM), a formal security evaluation method. The proposed scheme is also compared in terms of computation cost, communication cost and memory overhead against relevant existing schemes. The total computation cost of the proposed scheme is 7.76 ms, which is 8.14%, 5.20%, and 11.40% schemes. The results show that the proposed scheme is both efficient and secure, proving its viability.

## 1. Introduction

Micro Aerial Vehicles (MAVs) are getting a lot of attention from research organizations and businesses around the world [1]. These flying machines have proven their worth in situations where humans cannot reach or work efficiently, such as last-minute package delivery during rush hours or base searches in inaccessible areas of the battlefield. Compared to conventional methods, MAVs can significantly lower the risk to human life, increase the system’s efficiency, and shorten the time of operations. The broad capabilities of MAVs range from surveillance MAVs with fixed wings to advanced MAVs capable of hovering, navigation, carrying several sensors, and carrying out their missions up to several kilometers in range [2]. MAVs can transmit data to a portable base station and can exchange data with one another. A general architecture of MAVs network is depicted in Figure 1. Despite these benefits, MAVs are not suitable for real-time or processor-intensive applications because to their limited memory and processing power [3].

Apart from the aforementioned constraints, the security measures to fight against cyber-attacks are rarely considered during the design of MAVs [4]. The security and privacy of the network could be severely compromised due to this vulnerability, which would have a devastating effect on data transmission and storage. There are a variety of ways a malicious attacker can compromise the MAVs system. The malicious attacker can, for instance, send several reservation requests, eavesdrop on control messages, or fake data. Wi-Fi-connected MAVs are more vulnerable to cyber-attacks than cellular-connected ones because of their less-reliable connections and weaker security measures [5]. Tracking MAV locations, tampering with onboard hardware, illegal data access, message modification, and fabrication are examples of common privacy and security concerns across the MAV system [6,7]. A major security concern that compromises the privacy of MAVs is a Global Positioning System (GPS) spoofing attack [8,9,10], in which an attacker exploits GPS signals. In this method, an adversary sends an MAV slightly stronger GPS signals in order to deviate it from its original mission. Therefore, given their extensive usage in current military and commercial applications, there is an urgent need for enhanced security measures for MAVs.

Authentication and confidentiality are two of the most important aspects of any security protocol design for ensuring secure communication, and the same is applicable for MAVs security. Encryption and digital signatures provide solutions for confidentiality and authenticity respectively. When both attributes are required simultaneously and in a single logical step for devices with limited resources, such as MAVs, signcryption [11] is preferred. In addition, generalized signcryption is an extension of the signcryption scheme that not only offers encryption and digital signature simultaneously, but also has the option to offer both independently, if desired. Such a characteristic is useful if one of the two essential characteristics, confidentiality or authenticity, is desired [12]. Generalized signcryption can be used in ring configurations, known as ring signcryption, which offers advantageous characteristics such as anonymity, spontaneity, flexibility, and equal membership [13]. A conditional privacy preserving property can be implemented in addition to generalized ring signcryption to guarantee recipient and sender identify anonymity. In this approach, each entity encrypts their real identity using a common secret key between entity and PKG in the key generation process rather than using the real identities of sender and receiver. PKG must first locate the secret key and real identity after obtaining the encrypted identity. The encrypted identities of each user for signcryption and unsigncryption are then published by PKG.

Zhou et al. [14] proposed a concrete scheme for generalized ring signcryption in an identity-based framework. The proposed technique is based on bilinear pairing, and a random oracle model (ROM) is used for the security analysis. Due to the fact that the scheme [14] is based on bilinear pairing, which involves computationally expensive cryptographic operations, it is not suited for resource-constrained devices with low processing capabilities, such as MAVs, to conduct such operations. In addition, the proposed scheme lacks conditional privacy-preserving characteristics. Caixue Zhou [15] proposed a certificate-based generalized ring signcryption method and a concrete methodology employing bilinear pairings for certificate-based cryptosystems. Using the ROM, the security hardness of the proposed system is evaluated. Again, this scheme [15] is not suitable for MAVs due to the high computation cost of bilinear pairing and the absence of conditional privacy-preserving attribute.

M. Luo and Y. Zhou [16] introduced an efficient conditional privacy-preserving authentication protocol based on generalized ring signcryption scheme. Generalized ring signcryption is proposed in this protocol to provide ring signature mode and ring signcryption mode inside a single algorithm in order to meet the diverse security needs of complicated application scenarios. A practical public verification technique is meant to make tracking results verifiable and more trustworthy. In addition, the protocol accomplishes secrecy, immutability, and Known Session-Specific Temporary Information Security (KSSTIS). However, the proposed protocol involves bilinear pairing-based multiplication, modular exponentials, and bilinear pairing in the combined ring signature and signcryption method, which is incompatible for MAVs. Khan et al. [17] presented an identity-based generalized signcryption with multi-access edge computing option to protect Flying Ad hoc Networks (FANETs). However, neither conditional privacy preservation nor ring signcryption are supported by the proposed scheme. Consequently, this scheme [17] does not ensure anonymity. Din et al. [18] presented an improved identity-based generalized signcryption scheme for secure multi-access edge computing-enabled FANETs. The proposed scheme supports neither conditional privacy preservation nor ring signcryption. Therefore, this approach [18] does not guarantee anonymity.

With the aforementioned facts and favorable features in mind, we provide a conditional privacy-preserving generalized ring signcryption scheme for MAVs in this work. Moreover, the proposed scheme is based on an Identity-based public key cryptosystem, which uses the user’s name, IP address, etc. as his/her public key, hence eliminating the requirement for a public key certificate. Then, a trusted third party known as the PKG produces all users’ private keys, which introduces a new issue known as the private key escrow problem. However, it is still quite beneficial in situations when the PKG is completely trusted. The following are the main contributions of the proposed scheme that distinguish it from existing schemes.

We propose a conditional privacy-preserving generalized ring signcryption scheme for MAVs using the ECC operation.The proposed scheme is conditional privacy-preserving, meaning each entity encrypts its real identity using a common secret key between entity and PKG in the key generation process.The proposed scheme enables encryption and digital signature simultaneously as well as independently using generalized signcryption. In ring configurations mode, this scheme guarantees anonymity, spontaneity, flexibility, and equal membership.We conducted a formal security study using the Random Oracle Model (ROM) and found that the proposed scheme is secure against a wide range of cyber-attacks.Finally, the proposed scheme’s efficiency is compared to its counterparts, validating its low computation cost, communication cost and memory overhead.

The structure of the article is as follows: Section 2 provides preliminary information, the network model, and the syntax of the proposed scheme. In contrast, Section 3 includes a security analysis of the proposed scheme. In Section 4, performance analysis is discussed. The conclusion is contained in Section 5.

## 2. Preliminaries, Network Model and Syntax of the Proposed Scheme

This section includes preliminaries (elliptic curve cryptography, the elliptic curve decisional Diffie–Hellman problem, the elliptic curve discrete logarithm problem), syntax of the proposed scheme, network model and notations for the proposed scheme as shown in Table 1.

### 2.1. Preliminaries

#### 2.1.1. Elliptic Curve Cryptography (*ECC*)

Suppose $G_{ECC}$ is a finite cyclic group on the elliptic curve ($E_{ECC}$), $f_{q}$ is the finite field of $E_{ECC}$ with prime order q, let q>3, and ξ is the generator of group $G_{ECC}$; the elliptic can be defined as follows: $V^{2}=U^{3}+sU+t$ on $f_{q}$. Suppose $\left(U,V\right)\in f_{q}$ × $f_{q}$ based on the point, which is called infinity point on elliptic curve $\left(Ô\right)$ and congruence $V^{2}$ ≡ $U^{3}+sU+tmod\;q$, where the values $\left(s,t\right)\in f_{q}$ satisfying $4s^{3}+27t^{2}mod\;q$.

#### 2.1.2. Elliptic Curve Decisional Diffie-Hellman Problem (ECDDHP)

Assume ξ is the generator of group $G_{ECC}$ with prime order q, and given ($\Omega \cdot \xi ,\;\theta \cdot \;\xi ,\xi ,K\;\in G_{ECC}$), extracting θ and Ω from K=Ω·θ·ξ is called ECDDHP.

#### 2.1.3. Elliptic Curve Discrete Logarithm Problem (ECDLP)

Assume *ξ* is the generator of group G_*ECC*_ with prime order *q*, and given (*θ*.*ξ*,*ξ*,*K* ∈ G_*ECC*_), extracting *θ* from *K* = *θ*. *ξ* is called ECDLP.

### 2.2. Syntax

The syntax of the proposed scheme consists of the five sub-algorithms listed below. Initialization: The ground core network (GCN) can play the role private key generator (PKG), in which he/she can sets ${ß}_{GCN}$ as his/her secret key, $\delta_{GCN}$ as his/her public key, and generates a public parameter set Ж.Key Generation: The device that participates in a network as a legal user will send ($EId_{i},\;\Omega_{i}$) to GCN by using open channel. Based on ($EId_{i},\;\Omega_{i}$), GCN first compute $\gamma_{i}$ and recover the real identity $\;Id_{i}$. Then, GCN computes $\lambda_{i},$ $\Phi_{i}$ and send ($\Phi_{i},\;\lambda_{i}$) to the legitimate user by using secure channel.Generalized Ring Signcryption: This algorithm will run by Micro Aerial Vehicle (*MAV*), in which the *MAV* take input that are (${\mathrm{EId}}_{\mathrm{MAV}},m,\lambda_{X},{£}_{X},\delta_{GCN}$) and produce the tuple (κ,Л,Γ).Generalized Ring Signcryption Verifications: Given the tuple ($EId_{X},\lambda_{MAV},{£}_{MAV},\delta_{GCN},\kappa ,Л,\Gamma ,\Phi_{X}.$), a user can verify (κ,Л,Γ).

### 2.3. Network Model

Figure 2 depicts the network model of the proposed scheme, which includes entities such as MAVs and Base Station (BS) deployed to provide monitoring of a certain region. The proposed network model relies heavily on MAVs, which are outfitted with a camera, IMU, sensors, and GPS devices capable of handling a wide range of use cases. It allows for interaction between many MAVs and also between MAVs and fixed facilities. To establish a connection with the BS, the MAV makes use of 5G and Wi-Fi wireless technologies. The MAVs are able to talk to one another over Wi-Fi. The primary goal of a hybridised approach is to capitalise on the strengths of both technologies.

## 3. Proposed Scheme Construction

The construction of the proposed scheme includes the following steps.

Initialization: In this sub algorithm, a ground core network (GCN) can play the role private key generator (PKG) that can first choose his own secret key ${ß}_{GCN}\in f_{q}$ and compute a master public key as $\delta_{GCN}={ß}_{GCN}\cdot \xi$. then, GCN chooses three hash functions (${Ц}_{1}$*,*
${Ц}_{2},{Ц}_{3}$) that are irreversible and collision resistant. At the end, GCN produces a public param $Ж=(f_{q},\delta_{GCN},\xi ,\;{Ц}_{1}$*,*
${Ц}_{2},{Ц}_{3}$).

Key Generation: In this sub algorithm, a device which participated in a network as a legal user will send his encrypted real identity $EId_{i}=\gamma_{i}⊕Id_{i}$, and $\Omega_{i}=\alpha_{i}\cdot \xi ,\;$ to GCN by using open channel, where $\gamma_{i}=\alpha_{i}\cdot \delta_{GCN}$ and $\alpha_{i}\in f_{q}$. Based on ($EId_{i},\;\Omega_{i}$), GCN firs compute $\gamma_{i}={ß}_{GCN}\cdot \Omega_{i}$ and recover the real identity $Id_{i}$ as $Id_{i}=EId_{i}⊕\gamma_{i}$. Then, GCN choose $\eta_{i}\in f_{q}$, compute $\lambda_{i}=\eta_{i}\cdot \xi$*,*
${£}_{i}={Ц}_{1}\left(Id_{i},\lambda_{i}\right)$*,* calculate $\Phi_{i}=\eta_{i}+{£}_{i}\cdot {ß}_{GCN}$, and send ($\Phi_{i},\;\lambda_{i}$) to the legitimate user by using secure channel.

Generalized Ring Signcryption: This algorithm will run by MAV, in which the MAV first select his identity (${\mathrm{EId}}_{\mathrm{MAV}\;}$) from $\Delta =\{EId_{\mathrm{MAV}\;1}$*,*
$EId_{\mathrm{MAV}\;2},EId_{\mathrm{MAV}\;3},\ldots \ldots ,EId_{\mathrm{MAV}n}\}$ and perform the following steps.MDN choose $\chi_{\mathrm{MAV}\;}\in f_{q}$ and compute $Л=\chi_{\mathrm{MAV}\;}\cdot \xi$.Compute $\Psi =\chi_{\mathrm{MAV}\;}\left(\lambda_{X}+{£}_{X}\cdot \delta_{GCN}\right)$ *and*
$\Gamma ={Ц}_{2}\left(\Psi \right)⊕\left(m,EId_{MAV\;}\right).$Compute $\omega ={Ц}_{3}\left(EId_{MAV\;},\lambda_{MAV},\lambda_{X},Л,\Gamma \right)$ *and*
$\kappa =\chi_{\mathrm{MAV}\;}+\omega \cdot \Phi_{MAV}$.MAV send (ω,Л,Γ) to everything (*X*).


Generalized Ring Signcryption Verifications: With the encrypted identity ($EId_{X}$), a user upon reception of (ω,Л,Γ) can perform the following steps.Compare if $\kappa \cdot \xi =Л+\omega \cdot \left(\lambda_{MAV}+{£}_{MAV}\cdot \delta_{GCN}\right)$ is holds, where $\omega ={Ц}_{3}\left(EId_{MAV},\lambda_{MAV},\lambda_{X},Л,\Gamma \right)$.Compute $\Psi =\Phi_{X}\cdot Л$ and $\left(m,EId_{MAV}\right)=\Gamma ⊕{Ц}_{2}\left(\Psi \right)$.

### Correctness Analysis

The device at receiving end (*X*) can verify the signature as follows.
(1)$$\begin{array}{c} \kappa \cdot \xi =Л+\omega \cdot \left(\lambda_{MAV}+{£}_{MAV}\cdot \delta_{GCN}\right)=\left(\chi_{MAV}+\omega \cdot \Phi_{MAV}\right)\cdot \xi =(\chi_{MAV}\cdot \xi +\\ \omega \cdot \Phi_{MAV}\cdot \xi )=\left(\chi_{MAV}\cdot \xi +\omega \cdot \left(\eta_{MAV}+{£}_{MAV}\cdot {ß}_{GCN}\right)\cdot \xi \right)=(\chi_{MAV}\cdot \xi +\\ \omega \cdot \left(\eta_{MAV}\cdot \xi +{£}_{MAV}\cdot {ß}_{GCN}\cdot \xi \right))=\left(Л+\omega \cdot \left(\lambda_{MAV}+{£}_{MAV}\cdot \delta_{GCN}\right)\right) \end{array}$$
hence proved.

Furthermore, a device at receiving end (*X*) can made the decryption key as follows.
(2)$$\begin{array}{c} \Psi =\Phi_{X}\cdot Л=\left(\eta_{X}+{£}_{X}\cdot {ß}_{GCN}\right)\cdot \chi_{MAV}\cdot \xi =\left(\eta_{X}\cdot \xi +{£}_{X}\cdot {ß}_{GCN}\cdot \xi \right)\cdot \chi_{MAV}=\\ \left(\lambda_{X}+{£}_{X}\cdot \delta_{GCN}\right)\cdot \chi_{MAV}=\chi_{MAV}\left(\lambda_{X}+{£}_{X}\cdot \delta_{GCN}\right) \end{array}$$
hence proved.

## 4. Security Analysis

In this section, we first show that the proposed scheme is secure against breaches of confidentiality and forgeability under the Random Oracle Model (ROM). Then, using an informal security analysis, we show that the proposed scheme is secure against an adversary attempting to violate sender and recipient anonymity. The subsequent theorems demonstrate that the proposed scheme provides security properties such as confidentiality, unforgeability, sender anonymity, and recipient anonymity, respectively.

**Theorem** **1.** 
***Confidentiality:** The proposed generalized ring signcryption is indistinguishable against intruder INT under the ROM, if ECDDHP is hard.*


**Proof.** Suppose the instances of elliptic curve $\left(\Omega \cdot \xi ,\;\theta \cdot \;\xi ,\xi ,K\;\in G_{ECC}\right)$ is given to $C_{\mathrm{ECDDHP}}$. To find θ and Ω from K=Ω·θ·ξ, $C_{\mathrm{ECDDHP}}$ will play the following Game with INT.Initialization: $C_{\mathrm{ECDDHP}}$ can first choose the secret key ${ß}_{GCN}\in f_{q}$, public key $\delta_{GCN}$, public parameter set Ж. Then, $C_{\mathrm{ECDDHP}}$ sends Ж to INT.Phase 1: Here, INT can made the following queries with $C_{ECDDHP}.$
${Ц}_{1}$Query: INT send a request for ${Ц}_{1}$ Query with identity $(Id_{i})\;C_{ECDDHP}$ check for a tuple $\left(Id_{i},\lambda_{i},{£}_{i}\right)$ in the list $L_{{Ц}_{1}}$, if $\left(Id_{i},\lambda_{i},{£}_{i}\right)$ is found, $C_{ECDDHP}$ returns ${£}_{i}$ to INT. Otherwise, $C_{ECDDHP}$ choose the value for ${£}_{i}$ randomly and returns it to INT.${Ц}_{2}$Query: INT send a request for ${Ц}_{2}$ Query with identity $(Id_{i})\;C_{ECDDHP}$ check for a tuple $\left(\Psi_{i},{£}_{1i}\right)$ in the list $L_{{Ц}_{2}}$, if $\left(\Psi_{i},{£}_{1i}\right)$ is found, $C_{ECDDHP}$ returns ${£}_{1i}$ to INT. Otherwise, $C_{ECDDHP}$ choose the value for ${£}_{1i}$ randomly and returns it to INT.${Ц}_{3}$Query: INT send a request for ${Ц}_{3}$ Query with identity $(Id_{i})\;C_{ECDDHP}$ check for a tuple $\left(EId_{i},\lambda_{i},\Gamma_{i},{Л}_{i},\omega_{i}\right)$ in the list $L_{{Ц}_{3}}$, if $\left(EId_{i},\lambda_{i},\Gamma_{i},{Л}_{i},\omega_{i}\right)$ is found, $C_{ECDDHP}$ returns $\omega_{i}$ to INT. Otherwise, $C_{\mathrm{ECDDHP}}$ choose the value for $\omega_{i}$ randomly and returns it to INT.User Public Key Query: INT send a request for User Public Key Query with $(Id_{i},\lambda_{i}),\;C_{ECDDHP}$ check for a tuple $\left(Id_{i},\lambda_{i}\right)$ in the list $L_{UPK}$, if $\left(Id_{i},\lambda_{i}\right)$ is found, $C_{ECDDHP}$ returns $\lambda_{i}$ to INT. Otherwise, $C_{ECDDHP}$ perform the following two steps.At jth query, if i=j, $C_{ECDDHP}$ set $\lambda_{i}=\Omega \cdot \xi$.Else, compute $\lambda_{i}=\eta_{i}\cdot \xi$, where it selects $\eta_{i}$ randomly.At the end, $C_{ECDDHP}$ returns $\lambda_{i}$ to INT.User Private Key Query: INT send a request for User Private Key Query with $(Id_{i},\lambda_{i},\Phi_{i}),\;\;C_{ECDDHP}$ check for a tuple $\left(Id_{i},\lambda_{i},\Phi_{i}\right)$ in the list $L_{UPRK}$, if $Id_{i}=Id,$ $C_{ECDDHP}$ stop further processing, otherwise he found the tuple $\;\left(Id_{i},\lambda_{i},\Phi_{i}\right)\;$ and returns $\Phi_{i}$ to INT.Generalized Ring Signcryption Query: INT send a request for Generalized Ring Signcryption with m, $EId_{MAV}$ and $EId_{X}$, where $EId_{MAV}\;\in \Delta =\{EId_{\mathrm{MAV}\;1}$, $EId_{\mathrm{MAV}\;2},\;EId_{\mathrm{MAV}\;3},\ldots \ldots ,EId_{\mathrm{MAV}n}\}$ and $C_{ECDDHP}$ perform the following steps.If $EId_{MAV}\;!=Id$, It choose $\chi_{MAV}\in f_{q}$ and compute $Л=\chi_{MAV}\cdot \xi -\omega \left(\lambda_{MAV}+{£}_{MAV}\cdot \delta_{GCN}\right)$.Compute $\Psi =\chi_{MAV}\left(\lambda_{X}+{£}_{X}\cdot \delta_{GCN}\right)$ *and*
$\Gamma ={Ц}_{2}\left(\Psi \right)⊕\left(m,EId_{MAV}\right).$Compute $\omega ={Ц}_{3}\left(EId_{MAV},\lambda_{MAV},\lambda_{X},Л,\Gamma \right)$ *and*
$\kappa =\chi_{MAV}+y$, *where* y *is randomly selected now here.*$C_{ECDDHP}$ send (κ,Л,Γ) to INT.Generalized Ring Signcryption Verification Query: If $EId_{X}=Id$, $C_{ECDDHP}$ shows the tuple (κ,Л,Γ) is invalid. Otherwise, it normally Generalized Ring Signcryption Verification algorithm.Challenge: INT send the tuple ($m_{101}$, $m_{102},EId_{MAV},EId_{X}$) to $C_{ECDDHP}$, where $m_{101}$, $m_{102}$ are the two plaintexts with same size but contains different contents. If $EId_{X}=Id$, $C_{ECDDHP}$ pick $\iota \in \;\left\{0,\;1\right\}$ and perform the following computations.It computes Л=Ω·ξ.Compute $\Psi =K+{£}_{X}\cdot \delta_{GCN}\;$ *and*
$\Gamma ={Ц}_{2}\left(\Psi \right)⊕\left(m,EId_{MAV}\right).$Compute $\omega ={Ц}_{3}\left(EId_{MAV},\lambda_{MAV},\lambda_{X},Л,\Gamma \right)$ *and*
$\kappa =\omega \cdot \Phi_{MAV}+y+\Omega$*, where* y *is randomly selected now here.*$C_{ECDDHP}$ returns (κ,Л,Γ*).****Phase 2:*** In this phase, *INT* executes ${Ц}_{1}$ Query, ${Ц}_{2}$ Query, ${Ц}_{3}$ Query, User Public Key Query, Generalized Ring Signcryption Query, and Generalized Ring Signcryption Verification Query, respectively. Note that at this stage *INT* should not perform User Private Key Query on encrypted identity $EId_{X}$ and requested message corresponding to the Generalized ring signcrypted text.Guess: *INT* return $\iota^{/}\in \;\left\{0,\;1\right\}$, if $\iota =\iota^{/}$, $C_{ECDDHP}$ outputs 1. Otherwise, $C_{ECDDHP}$ outputs 0.Probability Analysis: Suppose $Q{Ц}_{1},Q{Ц}_{1},Q{Ц}_{1},Q_{UPK}$, and $Q_{UPRK}$ represent ${Ц}_{1}$ Query, ${Ц}_{2}$ Query, ${Ц}_{3}$ Query, User Public Key Query, and User Private Key Query, respectively. So, we express the following events.1.$\Theta_{1}:$ $C_{ECDDHP}$ succeeded in User Private Key Query.2.$\Theta_{2}:$ $C_{\mathrm{ECDDHP}}$ succeeded in Generalized Ring Signcryption Verification Query.3.$\Theta_{2}:$ $C_{\mathrm{ECDDHP}}$ succeeded in in challenge phase.After denoting the above events, we can easily receive the following outcomes.$\mathrm{Pr}\left(\Theta_{1}\right)=1-\frac{Q_{UPRK}}{Q_{UPK}},\;\mathrm{Pr}\left(\Theta_{2}\right)=1-\frac{1}{2^{j}},$ and $\mathrm{Pr}\left(\Theta_{3}\right)=\frac{1}{Q_{UPK}-Q_{UPRK}}$, then $\mathrm{Pr}\left(C_{ECDDHP}\;sucess\right)=\mathrm{Pr}\left(\Theta_{1}\wedge \Theta_{2}\wedge \Theta_{3}\right)=\mathrm{Pr}\left(\Theta_{1}\right)\cdot \mathrm{Pr}\left(\Theta_{2}\right)\cdot \mathrm{Pr}\left(\Theta_{3}\right)=\left(1-\frac{Q_{UPRK}}{Q_{UPK}}\right)\left(1-\frac{1}{2^{j}}\right)\left(\frac{1}{Q_{UPK}-Q_{UPRK}}\right)$≈ $\left(\frac{1}{Q_{UPK}}\right)$≈ $\frac{€}{Q_{UPK}}$, where € represent the advantage of *INT*. □

**Theorem** **2.** 
**Unforgeability.**
* Our proposed generalized ring signcryption is indistinguishable against intruder INT under the random oracle model, if ECDLP is hard.*


**Proof.** Suppose the instance of elliptic curve ($\Omega \cdot \xi ,\;\xi ,K\;\in G_{ECC}$) is given to $C_{\mathrm{ECDLP}}$ so, to find Ω from K=Ω·ξ, $C_{\mathrm{ECDLP}}$ will play the following Game with INT.Initialization: $C_{\mathrm{ECDLP}}$ can first choose the secret key ${ß}_{GCN}\in f_{q}$, public key $\delta_{GCN}$, public parameter set Ж. Then, $C_{\mathrm{ECDDHP}}$ send Ж to INT.Queries: All the queries are processed is same as executed in Theorem 1-Confidentiality.Forgery: INT wants to generate and verify combined ring signature and signcryption, in which he needs the private key of MAV and X ($\Phi_{MAV},\Phi_{X}$). INT can generate the forge signature as follows.INT choose $\chi_{INT}\in f_{q}$ and compute $Л=\chi_{INT}\cdot \xi$.Compute $\Psi =\chi_{INT}\left(\lambda_{X}+{£}_{X}\cdot \delta_{GCN}\right)$ *and*
$\Gamma ={Ц}_{2}\left(\Psi \right)⊕\left(m,EId_{MAV}\right).$Compute $\omega ={Ц}_{3}\left(EId_{MAV},\lambda_{INT},\lambda_{X},Л,\Gamma \right)$ *and*
$\kappa =\chi_{INT}+\omega \cdot \Phi_{INT}$.Returns (ω,Л,Γ).In the above process for forging a signature, *INT* can solve two-time ECDLP such as finding the values ($\chi_{MAV},\Phi_{MAV}$*).*Probability Analysis: Suppose $Q{Ц}_{1},\;Q{Ц}_{1},\;Q{Ц}_{1},Q_{UPK}$, and $Q_{UPRK}$ represent ${Ц}_{1}$ Query, ${Ц}_{2}$ Query, ${Ц}_{3}$ Query, User Public Key Query, and User Private Key Query, respectively. So, we express the following events.4.$\Theta_{1}:$ $C_{ECDDHP}$ succeeded in User Private Key Query.5.$\Theta_{2}:$ $C_{ECDDHP}$ succeeded in Generalized Ring Signcryption Verification Query.6.$\Theta_{2}:$ $C_{ECDDHP}$ succeeded in in challenge phase.After denoting the above events, we can easily receive the following outcomes.$\mathrm{Pr}\left(\Theta_{1}\right)=1-\frac{Q_{UPRK}}{Q_{UPK}},\;\mathrm{Pr}\left(\Theta_{2}\right)=1-\frac{1}{2^{j}},$ and $\mathrm{Pr}\left(\Theta_{3}\right)=\frac{1}{Q_{UPK}-Q_{UPRK}}$, then $\mathrm{Pr}\left(C_{ECDDHP}\;sucess\right)=\mathrm{Pr}\left(\Theta_{1}\wedge \Theta_{2}\wedge \Theta_{3}\right)=\mathrm{Pr}\left(\Theta_{1}\right)\cdot \mathrm{Pr}\left(\Theta_{2}\right)\cdot \mathrm{Pr}\left(\Theta_{3}\right)=\left(1-\frac{Q_{UPRK}}{Q_{UPK}}\right)\left(1-\frac{1}{2^{j}}\right)\left(\frac{1}{Q_{UPK}-Q_{UPRK}}\right)$≈ $\left(\frac{1}{Q_{UPK}}\right)$≈ $\frac{€}{Q_{UPK}}$, where € represents the advantage of *INT*. □

**Theorem** **3.** **Sender Anonymity.*** In the key generation phase, the sender device called MAV will send his encrypted real identity*$EId_{MAV}=\gamma_{MAV}⊕Id_{MAV}$*, and*$\Omega_{MAV}=\alpha_{MAV}\cdot \xi ,\;$*to GCN by using open channel, where*$\gamma_{MAV}=\alpha_{MAV}\cdot \delta_{GCN}$ *and*$\alpha_{MAV}\in f_{q}$. *Based on (*$EId_{MAV},\;\Omega_{MAV}$*), GCN firs compute*$\gamma_{MAV}={ß}_{GCN}\cdot \Omega_{MAV}$ *and recover the real identity*$Id_{MAV}$*as*$Id_{MAV}=EId_{i}⊕\gamma_{MAV}$*. Here, if INT wants the real identity*$Id_{MAV}$*of MAV, he will pass the following two cases.*
INT first struggle to access $\alpha_{MAV}$ from $\Omega_{MAV}=\alpha_{MAV}\cdot \xi$ to made $\gamma_{MAV}=\alpha_{MAV}\cdot \delta_{GCN}$.Secondly *INT* can access ${ß}_{GCN}$ from $\delta_{GCN}={ß}_{GCN}\cdot \xi$ to made $\gamma_{MAV}={ß}_{GCN}\cdot \Omega_{MAV}$.
In both the above cases, *INT* can solve *ECDLP* which will be infeasible for him/her.

**Theorem** **4.****Receiver Anonymity.*** In the key generation phase, the receiver device called*X *will send his encrypted real identity*$EId_{X}=\gamma_{X}⊕Id_{X}$*, and*$\Omega_{X}=\alpha_{X}\cdot \xi ,\;$*to GCN by using open channel, where*$\gamma_{X}=\alpha_{X}\cdot \delta_{GCN}$ *and*$\alpha_{X}\in f_{q}$*. Based on (*$EId_{X},\;\Omega_{X}$*), GCN firs compute*$\gamma_{X}={ß}_{GCN}\cdot \Omega_{X}$*and recover the real identity*$Id_{X}$*as*$Id_{X}=EId_{X}⊕\gamma_{X}$*. Here, if INT wants the real identity*$Id_{X}$ *of*X*, he will pass the following two cases.*
INT first struggle to access $\alpha_{X}$ from $\Omega_{X}=\alpha_{X}\cdot \xi$ to made $\gamma_{X}=\alpha_{X}\cdot \delta_{GCN}$.Secondly INT can access ${ß}_{GCN}$ from $\delta_{GCN}={ß}_{GCN}\cdot \xi$ to made $\gamma_{X}={ß}_{GCN}\cdot \Omega_{X}$.
In both the above cases, *INT* can solve *ECDLP,* which will be infeasible for him/her.

## 5. Performance Comparison

This section compares the performance of the proposed scheme with the relevant existing counterparts proposed by Zhou et al. [14], Zhou et al. [15], and Luo and Zhou [16].

### 5.1. Computation Cost

The computation cost represents the operational expenses spent by each user during the proposed generalized ring signcryption process and existing comparable schemes proposed by Zhou et al. [14], Zhou et al. [15], and Luo and Zhou [16]. In Table 2, we list the key operations of the proposed scheme, including Elliptic Curve Point Multiplication ($ECC_{PM}$), Bilinear Pairing Based Multiplication ($BP_{BM}$), Modular Exponentials ($MD_{EXP}$), and Bilinear Pairing ($BP_{OP}$). Table 3 contains the operating expenses, measured in milliseconds (ms), for the proposed scheme, as well as those of Zhou et al. [14], Zhou et al. [15], and Luo and Zhou [16]. The time requires for a single $ECC_{PM}$ takes 0.97 ms, $BP_{BM}\;,\;\;4.31\;\mathrm{ms},MD_{EXP}\;,\;\;1.25\;\mathrm{ms}$ and $BP_{OP}\;\mathrm{takes}\;14.90$ [19]. The Multi-Precision Integer and Rational Arithmetic C Library (MIRACL) [20] is used to assess the performance of the proposed scheme by testing the runtime of the core cryptographic operations up to 1000 times. Observations are made on a workstation with the following specifications: 8 GB RAM and the Windows 7 Home Basic 64-bit operating system [21]. As seen in Figure 3, the proposed scheme has a lower computation cost than the schemes proposed by Zhou et al. [14], Zhou et al. [15], and Luo and Zhou [16].

### 5.2. Communication Cost

In this subsection, the proposed scheme is compared to existing schemes, namely those proposed by Zhou et al. [14], Zhou et al. [15], and Luo and Zhou [16], in terms of communication cost. We list the communication cost incurred based on the Elliptic Curve Parameter Size (*|ECC q|*), Bilinear Pairing Parameter Size (*|BP G|*)*,* and a message size (*|m|*) for the proposed and those of Zhou et al. [14], Zhou et al. [15], and Luo and Zhou [16]. We have selected the bit values 160, 1024, and 1024 bits for (*|ECC q|*), (*|m|*), and (*|BP G|*) from [19]. In addition, the communication cost analysis between Zhou et al. [14], Zhou et al. [15], and Luo and Zhou [16] and the proposed scheme are provided in Table 4. As seen in Figure 4, the proposed scheme has a lower communication cost than the schemes proposed by Zhou et al. [14], Zhou et al. [15], and Luo and Zhou [16].

### 5.3. Memory Overhead

The proposed scheme is compared in terms of memory overhead to existing schemes proposed by by Zhou et al. [14], Zhou et al. [15], and Luo and Zhou [16]. Table 5 describes the primary operations, and Table 6 compares the memory overhead in bits of the proposed scheme to that of relevant existing schemes. A significant reduction in memory space is achieved, as shown in Figure 5.

## 6. Conclusions

In this article, we proposed a conditional privacy-preserving generalized ring signcryption scheme for MAVs using an identity-based cryptosystem. The proposed scheme is developed using the Elliptic Curve Cryptography concept (ECC). A comprehensive security analysis of ROM indicates that the proposed method is robust to a number of attacks. Comparing the proposed scheme to similar schemes presented by Zhou et al. [14], Zhou et al. [15], and Luo and Zhou [16] with regard to commutation and communication costs. The results reveal that the proposed scheme is more cost-effective in terms of computation and communication costs than its current alternatives. In addition, the results demonstrate that the proposed method is suitable for MAV systems due to the algorithm’s functionality and reduced computation cost, communication cost and memory overhead.

## Figures and Tables

**Figure 1 micromachines-13-01926-f001:**
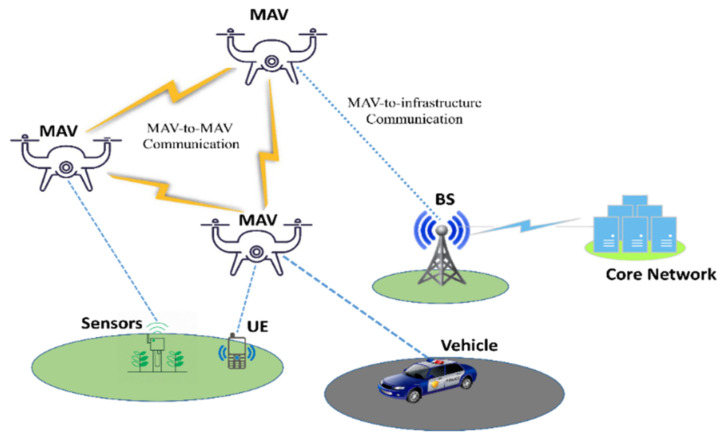
General architecture of MAVs network.

**Figure 2 micromachines-13-01926-f002:**
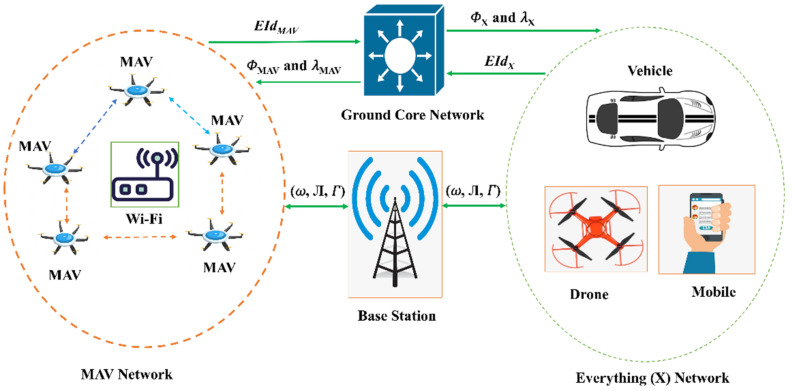
Network model of the proposed scheme.

**Figure 3 micromachines-13-01926-f003:**
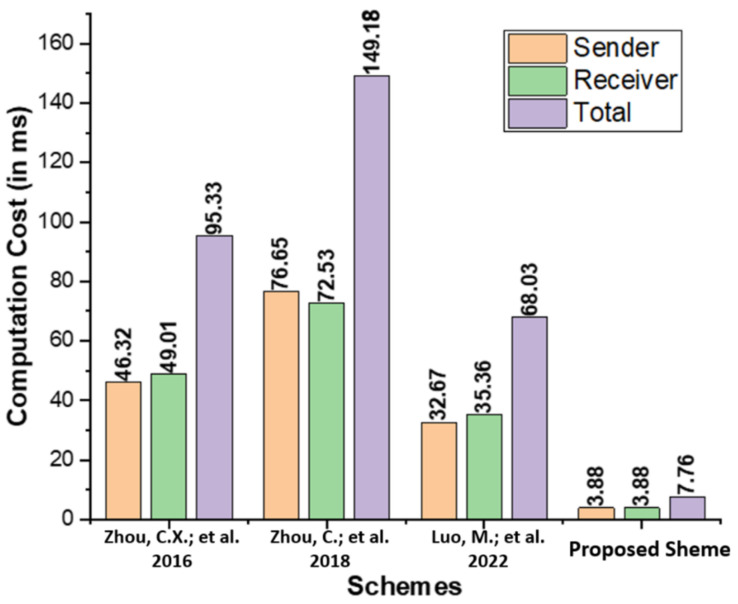
Comparison of computation cost (in ms) [14,15,16].

**Figure 4 micromachines-13-01926-f004:**
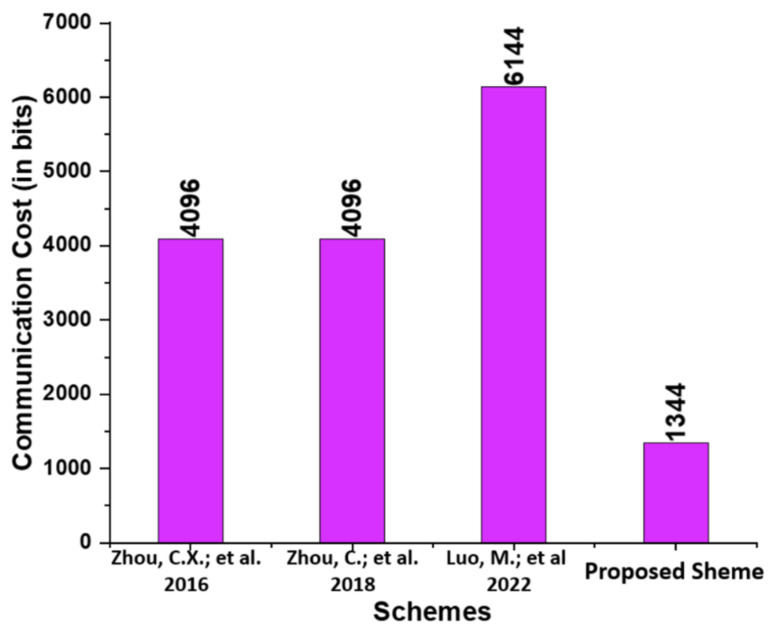
Comparison of communication cost (in bits) [14,15,16].

**Figure 5 micromachines-13-01926-f005:**
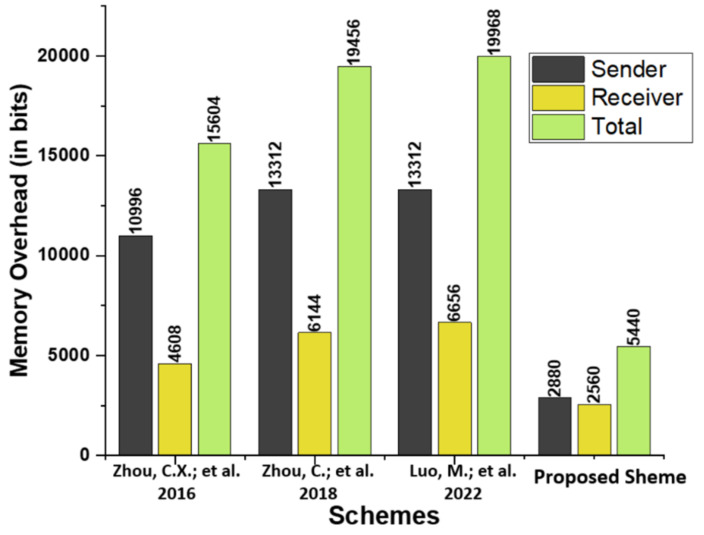
Comparison of memory overhead (in bits) [14,15,16].

**Table 1 micromachines-13-01926-t001:** Notation table.

S. No	Notation	Descriptions
1	GCN	Ground core network
2	PKG	Private key generator
3	Ж	Public parameter param
4	${Ц}_{1}$ *,* ${Ц}_{2},{Ц}_{3}$	Irreversible and collision resistant hash functions
5	$\delta_{GCN}$	Master secret key of ground core network
6	$\delta_{GCN}$	Master public key of ground core network
7	ξ	Generator of group $G_{ECC}$
8	$G_{ECC}$	Finite cyclic group on the elliptic curve $E_{ECC}$
9	$E_{ECC}$	The elliptic curve defined on $V^{2}=U^{3}+sU+t$
10	$EId_{MAV}$	Encrypted identity of MAV
11	MAV	It represents a Micro Aerial Vehicle (MAV)
12	$EId_{X}$	Encrypted identity of everything (X)
13	$Id_{MAV}$	Real identity of MAV
14	$Id_{X}$	Real identity of everything (X)
15	$f_{q}$	Finite field on the elliptic curve $E_{ECC}$ of order q
16	$\Phi_{MAV}$	Private key of MAV
17	$\Phi_{X}$	Private key of everything (X)
18	$\lambda_{X}$	Public key of everything (X)
19	$\lambda_{MAV}$	Public key of MAV
20	Δ	Identities of ring group $\{EId_{MAV\;1}$*,* $EId_{\mathrm{MAV}\;2},EId_{\mathrm{MAV}\;3},\ldots \ldots ,EId_{\mathrm{MAV}n}\}$
21	$\gamma_{MAV}$	Encryption and decryption key for real identity of MAV
22	$\gamma_{X}$	Encryption and decryption key for real identity of everything (X)
23	Ψ	Encryption and decryption key for message MAV and everything (X)
24	⊕	Used for Encryption and decryption

**Table 2 micromachines-13-01926-t002:** Comparison of computation cost with major operations.

Schemes	Sender	Receiver	Total
Zhou et al. [14]	$7BP_{BM}+1MD_{EXP}+1BP_{OP}$	$1BP_{BM}+3BP_{OP}$	$8BP_{BM}+1MD_{EXP}+4BP_{OP}$
Zhou et al. [15]	$10BP_{BM}+3MD_{EXP}+2BP_{OP}$	$3BP_{BM}+4BP_{OP}$	$13BP_{BM}+3MD_{EXP}+6BP_{OP}$
Luo and Zhou [16]	$7BP_{BM}+2MD_{EXP}$	$1BP_{BM}+1MD_{EXP}+2BP_{OP}$	$8BP_{BM}+3MD_{EXP}+2BP_{OP}$
Proposed Scheme	$4ECC_{PM}$	$4ECC_{PM}$	$8ECC_{PM}$

**Table 3 micromachines-13-01926-t003:** Comparison of computation cost (in ms).

Schemes	Sender	Receiver	Total
Zhou et al. [14]	7×4.31+1×1.25+1×14.9=46.32	1×4.31+3×14.90=49.01	8×4.31+1×1.25+4×14.90=95.33
Zhou et al. [15]	10×4.31+3×1.25+2×14.90=76.65	3×4.31+4×14.90=72.53	13×4.31+3×1.25+6×14.90=149.18
Luo and Zhou [16]	7×4.31+2×1.25=32.67	1×4.31+1×1.25+2×14.90=35.36	8×4.31+3×1.25+2×14.90=68.03
Proposed Scheme	4×0.97=3.88	4×0.97=3.88	8×0.97=7.76

**Table 4 micromachines-13-01926-t004:** Comparison of communication cost (in bits).

Schemes	Communication Cost	Communication Cost in Bits
Zhou et al. [14]	$\left|m\right|+3|BP_{G}|$	$\left|1024\right|+3\times \left|1024\right|=4096$
Zhou et al. [15]	$\left|m\right|+3|BP_{G}|$	$\left|1024\right|+3\times \left|1024\right|=4096$
Luo and Zhou [16]	$\left|m\right|+5|BP_{G}|$	$\left|1024\right|+5\times \left|1024\right|=6144$
Proposed Scheme	$\left|m\right|+2|ECC_{q}|$	$\left|1024\right|+2\times \left|160\right|=1344$

**Table 5 micromachines-13-01926-t005:** Memory Overhead Analysis.

Schemes	Sender	Receiver	Total
Zhou et al. [14]	9|$BP_{G}|+3\left|H\right|$+ $\left|m\right|$	3|$BP_{G}|+2\left|H\right|$+ $\left|m\right|$	12|$BP_{G}|+5\left|H\right|$+ $2\left|m\right|$
Zhou et al. [15]	11|$BP_{G}|+4\left|H\right|$+ $\left|m\right|$	4|$BP_{G}|+4\left|H\right|$+ $\left|m\right|$	15|$BP_{G}|+8\left|H\right|$+ $2\left|m\right|$
Luo and Zhou [16]	11|$BP_{G}|+4\left|H\right|$+ $\left|m\right|$	5|$BP_{G}|+2\left|H\right|$+ $\left|m\right|$	16|$BP_{G}|+6\left|H\right|$+ $2\left|m\right|$
Proposed Scheme	10|$ECC_{q}|+1\left|H\right|$+ $\left|m\right|$	8|$ECC_{q}|+1\left|H\right|$+ $\left|m\right|$	18|$ECC_{q}|+2\left|H\right|$+ $2\left|m\right|$

Note: |$ECC_{q}|=160,\left|H\right|=256,$|$BP_{G}|=1024$, and $\left|m\right|=1024$.

**Table 6 micromachines-13-01926-t006:** Memory Overhead Analysis in Bits.

Schemes	Sender	Receiver	Total
Zhou et al. [14]	9$\left|1024\left|+3\right|256\right|$+ $\left|1024\right|=10996$	3|$1024\left|+2\right|256|$+ $\left|1024\right|=4608$	12|$1024\left|+5\right|256|$+ $2\left|1024\right|=15604$
Zhou et al. [15]	11|$1024\left|+4\right|256|$+ $\left|1024\right|=13312$	4|$1024\left|+4\right|256|$+ $\left|1024\right|=6144$	15|$1024\left|+8\right|256|$+ $2\left|1024\right|=19456$
Luo and Zhou [16]	11|$1024\left|+4\right|256|$+ $\left|1024\right|=13312$	5|$1024\left|+2\right|256|$+ $\left|1024\right|=6656$	16|$1024\left|+6\right|256|$+ $2\left|1024\right|=19968$
Proposed Scheme	10|$160\left|+1\right|256|$+ $\left|1024\right|=2880$	8|$160\left|+1\right|256|$+ $\left|1024\right|=2560$	18|$160\left|+2\right|256|$+ $2\left|1024\right|=5440$

## Data Availability

Not applicable.

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
