# Peer review of "A Conditional Privacy Preserving Generalized Ring Signcryption Scheme for Micro Aerial Vehicles"

_micromachines, 2022, doi:10.3390/mi13111926_

Round 1

Reviewer 1 Report

The current manuscript proposes a conditional privacy-preserving generalized ring signcryption scheme for MAVs using an identity-based cryptosystem Elliptic Curve Cryptography (ECC). 

1) In the abstract should mention how it can improve the previous system;

2) Line 90-101 use other paragraph format;

3) Section 2 Related work should be in the section 1, to do the survey for the relative work.

4) Except the scheme in Figure 1, can add one more figure to explain such model for better understanding. 

5) Table 1 can be the appendix, do not need to list in the main content.

6) Only two results in Figure 2 and 3 are not enough. Can add more results for such new method. 

Author Response

Dear Reviewer,

Please find our response letter in the attached document.

Thank you

Reviewer 2 Report

Dear Authors,

The article is about very up-to-date and interesting content. It demonstrates quantitatively the parameters characteristic of several encryption schemes. While the methods themselves are presented in detail, the article lacks a broader presentation of the sample of encrypted data. Is it limited to only one set of test data? I did not find in the manuscript an answer to the question of how the encryption performance is influenced by, for example, changing the length of the test data or changing its content. Please explain in this regard and / or supplement the article with this content.

Author Response

Dear Reviewer, 

Please find our revised draft and response letter in the attached document. 

Thank you!

Round 2

Reviewer 1 Report

Accepted as it is

Reviewer 2 Report

Dear Authors,

Thank you for making corrections in line with my comments. I accept the article in this form.
